# A Framework of State Estimation on Laminar Grinding Based on the CT Image–Force Model

**DOI:** 10.3390/s25010238

**Published:** 2025-01-03

**Authors:** Jihao Liu, Guoyan Zheng, Weixin Yan

**Affiliations:** 1State Key Laboratory of Ocean Engineering, School of Ocean and Civil Engineering, Shanghai Jiao Tong University, Shanghai 200240, China; 2Institute of Medical Robotics, School of Medical Engineering, Shanghai Jiao Tong University, Shanghai 200240, China; 3Institute of Robotics, School of Mechanical Engineering, Shanghai Jiao Tong University, Shanghai 200240, China; xiaogu4524@sjtu.edu.cn

**Keywords:** state estimation, CT image, milling force prediction

## Abstract

It is a great challenge for a safe surgery to localize the cutting tip during laminar grinding. To address this problem, we develop a framework of state estimation based on the CT image–force model. For the proposed framework, the pre-operative CT image and intra-operative milling force signal work as source inputs. In the framework, a bone milling force prediction model is built, and the surgical planned paths can be transformed into the prediction sequences of milling force. The intra-operative milling force signal is segmented by the tumbling window algorithm. Then, the similarity between the prediction sequences and the segmented milling signal is derived by the dynamic time warping (DTW) algorithm. The derived similarity indicates the position of the cutting tip. Finally, to overcome influences of some factors, we used the random sample consensus (RANSAC). The code of the functional simulations has be opened.

## 1. Introduction

In spinal surgeries, it is significantly risky to undertake mechanical operations, including bone drilling and milling [1]. In general, bone milling is used for bone removal to relieve compression on the spinal cord [2,3]. The surgeons have to perform complex operations in a narrow surgical field, and prevent a cutting tip from damaging soft tissues, such as nerves, blood vessels, and ligaments. It is extremely high dependent on the surgeons’ experience [4,5].

The image-based navigation system has been developed for surgeons to conduct surgical planning through patient-to-image registration [6,7]. It integrates with a stereotaxy device, which can track the surgical tools with markers [8]. It is critical for a safe surgery to determine the position of the cutting tip. However, it is impossible to not only sense the deflection of a cutting bit but also to obtain the position of the cutting tip [9].

To address this problem, many researchers have been developing state recognition methods [10]. Some methods employ the intra-operative signal to identify the bone state during cutting. These signals consist of the cutting force/torque [11], sound pressure [12], vibration [13], and motor power of a cutting tool [14]. The bone state is recognized by detecting the signal feature of the stepwise breakthrough in the time domain. Qu et al. [15] developed a backpropagation neural network for state recognition on vertebral laminar grinding. Four intra-operative signals, including the characteristic milling force, milling speed, milling depth, and ultrasonic scalped power, were loaded to detect the state of bone breakthrough. Jiang et al. [16] proposed an analytical force model to estimate the cutting depth. Its accuracy was up to 0.2 mm, and this performance was dependent upon identification of the force coefficients inside the milling force model. Most of the intra-operative signals are affected by the heat released by the bone cutting, except for the vibration signal. Xia et al. [17] developed a vibration signal fusion method to predict the remaining thickness of the lamina in real time, from which, the lamina cutoff had a success rate of 98.4%. There can be variations in the bone cutting states and signal features for different specimens because those are functions of bone tissue properties [18,19,20]. Therefore, there existed a potential risk for the above methods from the perspectives of robustness and adaption.

The bone cutting force can be predictable. The mechanical model of the cutting force has been introduced from the field of metal manufacturing into the field of medical engineering [21,22,23]. In the mechanical model of the bone cutting, some force coefficients are different due to a change in the yield strength of bones. It is only through complex experiments that the force coefficients for the precious prediction on the drilling force can be calibrated [24].

Researchers have been focusing on the challenge of how to obtain some prior information and estimate the surgical states before an operation. The pre-operative CT image has been used to predict the cutting state [25]. CT images can digitize the bone shape and material properties well. Grayscale CT images can quantify the strength of the bone [26,27]. Williamson et al. [28] found that the drilling force sequence was similar to the grayscale array of the planning path extracted from the CT image. Therefore, Wang et al. [6] combined the pre-operative CT image and the intra-operative drilling force signal to estimate the drilling position. However, the accuracy was affected by the sampling synchronization of the drilling signal. Thus, the CT image is combined with the mechanical model of the drilling force to derive the CT image–force mapping model. Based on this model, Li et al. [25] developed the virtual sensing framework to predict the trend of the thrust change during the drilling process. The discrimination capacity of identification can achieve the voxel level of the CT images. According to the gray value array in the CT image subject to the surgical planning, Li et al. [29] proposed a strategy to adjust the cutting speed. Despite the existing models considering the mechanism between a cutting tool and bone tissue, and from the results of these published works, the prior force information derived from the CT images does not seem to be consistent with actual measurements [6,29].

The focus of this work is on estimating the position of the cutting tip, rather than directly avoiding injury to tissues. The relative code can be found at https://github.com/GHow-sjtu/LaminarStateEstimator-CT. Based on the existing knowledge of the CT image–force mapping model, we propose a method to estimate the real-time position of the cutting tip. The main contributions lie in three aspects:We develop an intra-operative state estimation for laminar grinding. The pre-operative CT image and the intra-operative milling force signal are combined to estimate the tip position. Simulation results show that the proposed estimation method is robust for different bones from various specimens, and it can adopt CT images from different centers.We establish a CT image–force model to generate prior knowledge of the bone milling state before an operation. The proposed method may not provide an accurate quantitative estimation of the milling force due to various bone properties. However, the prior knowledge contains the time-domain signal characteristics that are used to estimate the milling state.Experiments showed that the milling force sequence is relative to the density distribution along one milling path. The bone density distribution of one milling path in a lamina is independent. It can work as one of the identification features for positioning during cutting. It means that the proposed framework may no longer be effective when the object bone becomes brittle due to osteoporosis or degeneration.

Despite the strengths of our study, we acknowledge that there are certain limitations. We ignore the impact of the sampling frequency of the cutting force sensor in this work. We believe that addressing this limitation in future studies will provide a more robust method for clinical application. Moreover, the effectiveness of the proposed method depends on the characteristics of bone density distribution. Although we have conducted many tests, we cannot determine the specific conditions under which the proposed method becomes ineffective.

The paper is organized as follows. Section 2 presents the proposed method. Section 3 describes the experiments and results. Finally, we present discussions in Section 4, followed by our conclusion in Section 5.

## 2. Methods

### 2.1. Method Overview

In this work, a framework is developed to estimate the tip position during the operation process, as shown in Figure 1. The proposed framework adopts the pre-operative CT images and real-time milling signal as the inputs, and it consists of five functional modules. Module 1 extracts the CT voxels according to the surgical planning. Module 2 utilizes the CT image–force model to predict milling sequences in view of the thrust force, torque, and lateral forces. Module 3 segments the milling force signal into four-channel sequences by a tumbling window. The above data flows are combined into Module 4 to estimate the instantaneous tip position. The fitting algorithms in Module 5 are used to eliminate the nonlinear disturbances, as well as to estimate the position of the milled part. Finally, we can obtain the estimations on the position of the cutting tip, as well as the deviation of the actual milled paths.

The CT image–force model in Module 2 will be introduced in Section 2.2. This model is derived based on the mechanical model of ball-end milling. The tip position’s estimation in Module 4 will be detailed in Section 2.3, and the fitting algorithms in Module 2 will be proposed in Section 2.4.

In this work, because the cutter–workpiece engagement (CWE) is considered in the milling force model, we can ignore the impact of the milling strategy. As shown in Figure 2, a common layer-by-layer policy is used as the milling strategy [17]. The planning paths are filled in a rectangular space. By using this policy, the milling cutter will remove the bone tissue through multi-layer horizontal movements. A Cartesian coordinate system is established to model actions of the milling cutter. The *x*-axis is defined by the direction of the feed rate of the cutter. The *z*-axis is defined by the direction of the milling depth. The *y*-axis is defined according to the right-hand rule. And the origin is defined by the starting position of the surgical task.

### 2.2. Ct Image–Force Mapping

The CT image–force model allows translation of the high-dimensional 3D images into several-dimensional virtual force sequences. The instantaneous milling force results from the contact force at the cutting edges engaging with the bone. The contact force at the micro-edge varies with the bone strength at the contacting location. Because the bone strength is quantified by the gray value in the CT image, the bone milling process can be regarded as the interaction process of the cutting edges with the corresponding image voxels of the object bone. Thus, the contact force of a micro-edge can be modeled by **Equation** (Equation 1).
(1)$$f^{i}\left(t\right)=func\left(I\left(P_{i}\left(t\right)\right)\right)$$
where $f^{i}\left(t\right)$ indicates the prediction of the instantaneous force at the *i*-th micro-edge at time *t*, $P_{i}\left(t\right)$ indicates the position of the *i*-th micro-edge at time *t*, $I\left(\;\cdot \;\right)$ indicates the extracted image gray value at the corresponding position, and $func\left(\;\cdot \;\right)$ indicates the milling force in a function of the gray value.

It is noteworthy that the proposed CT image–force model is inaccurate when predicting the milling force due to many factors that are mentioned above. However, the predicted sequences encompass the same time-domain signal features.

#### 2.2.1. Geometric Model of the Ball-End Cutter

For modeling of the milling process of a ball-end cutter, we establish four coordinate systems, including the milling cutter coordinate system (MCCS), the instantaneous milling cutter coordinate system (*i*-MCCS), the local working coordinate system (*l*-WCS), and the workpiece coordinate system (WCS). As shown in Figure 3, these four coordinate systems are used to define the geometric model of cutting edges, instantaneous cutting force, and milling motions.

The MCCS, *i*-MCCS, and *l*-WCS are all defined at the ball-end cutter, and their origins are located at the centroid of the ball end, while the workpiece coordinate system (WCS) is defined at the object bone. The *x*-axis of the *i*-MCCS, *l*-WCS, and WCS is parallel to the feed direction of the milling cutter. The *z*-axis of the MCCS and *i*-MCCS is defined by the rod axis of the cutter. The *z*-axis of the *l*-WCS and WCS is normal to the milling layer. The *y*-axes of theses four coordinate systems are produced by the right-hand rule. During the milling process, there is an incline angle between the rob axis of the cutter and the bone surface.

In **Equation** (Equation 1), the coordinate position $P_{i}\left(t\right)$ of a micro-edge is one of the indispensable parameters for the milling prediction function. In order to obtain the position $P_{i}\left(t\right)$, the morphological and kinematic models of the ball-end cutter are built.

The morphological model of a ball-end cutter defines the position of cutting edges. In this work, a widely used orthopedic cutter with four flutes is chosen as the surgical tool. As shown in Figure 4, the position of the micro-edges on a ball-end cutter is modeled in the MCCS [21], as:
(2)$$\left\{\begin{array}{c} x_{m}=R\cos\theta_{m}\cos\left(\varphi_{0}-\sin\theta_{m}\tan\beta_{G}\right)\\ y_{m}=R\cos\theta_{m}\sin\left(\varphi_{0}-\sin\theta_{m}\tan\beta_{G}\right)\\ z_{m}=-R\sin\theta_{m} \end{array}\right.$$
where *R* is the radius of the ball end; $\varphi_{0}$ is the circumferential starting angle that defines the starting point of the cutting edge; $\beta_{G}$ is the helical angle of the cutting edge, $\beta_{G}=30^{\circ }$; and $\theta_{m}$ is the axial position angle of one micro-edge. Some cutting edges do not pass the tool center at the apex. As shown in Figure 4a, *h* is the distance from the edge to the center in the bottom view. Thus, the corresponding range of the axial position angle is within $\left[0,\arccos\left(h/R\;\right)\right]$.

#### 2.2.2. Kinematic Model of the Ball-End Cutter

As the milling cutter grinds, the interactions of cutting edges with the bone occur on a complex helix trajectory. The real-time position of the micro-edges in the *l*-WCS can be obtained through a spatial transformation, which is written as:(3)PW=Tt  W·Pt
where Tt  W is the transformation matrix from the milling cutter coordinate system (MCCS) to the workpiece coordinate system (WCS), which indicates the grinding motion of the milling cutter, as follows:(4)Tt  W=Tl  W·Ωtvt01
where $\Omega \left(t\right)$ denotes the self-spinning motion of the cutter, $v\left(t\right)$ denotes the feed motion, and Tl  W is the transformation matrix from the instantaneous milling coordinate system (*i*-MCS) to the local working coordinate system (*l*-WCS). It is defined by the incline angles $\delta_{mt}$ of the spindle axis in the workpiece coordinate system (WCS).
(5)Tl  W=10000cosδmt−sinδmt00sinδmtcosδmt00001

The spinning matrix $\Omega \left(t\right)$ and the motion vector $v\left(t\right)$ are modeled in the coordinate system *i*-MCCS, as follows:(6)$$\Omega \left(t\right)=\left(\begin{array}{ccc} \cos\left(\omega t+\phi_{0}\right) & -\sin\left(\omega t+\phi_{0}\right) & 0\\ \sin\left(\omega t+\phi_{0}\right) & \cos\left(\omega t+\phi_{0}\right) & 0\\ 0 & 0 & 1 \end{array}\right)$$
(7)$$v\left(t\right)={\left(\begin{array}{ccc} v_{d}\;\cdot \;t & 0 & 0 \end{array}\right)}^{T}$$
where ω is the self-spinning speed [rad/s], and $v_{d}$ is the feed rate [mm/s].

#### 2.2.3. Cutter–Workpiece Engagement (CWE)

Cutter–workpiece engagement (CWE) relates to the section of the cutter interacting with the bone. During the bone milling process, the salient feature of the ball-end milling is that the geometry and material properties of the surgical object vary along the milling path. It leads to the continuous change in the section of cutting edges participating in milling, as shown in Figure 5. When the micro-edges are inside the area of CWE, they are valid for the milling interaction. Thus, the calculation of CWE is the precondition to obtaining the milling prediction [21,23].

The CWE area can be modeled by the boundary curves. The boundary of the CWE area is an intersection between the envelope surface formed by the previous machined path and the semi-spherical surface of the ball-end cutter. Due to the usage of a ball-end cutter, the previous machined surface is regarded as a nonstandard cylindrical surface. As shown in Figure 5c, the CWE area is relative to the milling depth *d* and the step distance *w* among the milling paths. As shown in Figure 6, there exist five types of CWE areas during the layer-by-layer milling process. These figures are created by Function *mesh* and *surf* in the software *MATLAB 2021b*. The detailed equations that model the boundary curves of CWE areas are listed in Appendix A.

#### 2.2.4. Mechanical Model of Ball-End Milling

Lee and Altintas [22] propose the micro-element milling force model for orthogonal cutting, where the shearing mechanism and the plowing mechanism are considered separately. The formula that expresses the cutting forces at a micro-edge is shown in **Equation** (Equation 8).
(8)$$\left\{\begin{array}{c} dF_{r}=K_{rc}t_{n}db+K_{re}ds\\ dF_{a}=K_{ac}t_{n}db+K_{ae}ds\\ dF_{t}=K_{tc}t_{n}db+K_{te}ds \end{array}\right.$$
where $dF_{r}$, $dF_{a}$, and $dF_{t}$ are the radial, axial, and tangential forces at the micro cutting edge [N]; $K_{rc}$, $K_{ac}$, and $K_{tc}$ are the shear coefficients [N/mm]; $K_{re}$, $K_{ae}$, and $K_{te}$ are the blade force coefficients, and also, namely, edge specific coefficients [N/mm]; $t_{n}$ is the thickness of the undeformed chip [mm]; and db is the projection width of the micro-edge on the generatrix (the infinitesimal length of cutting edge) [mm], and is the projection length of the micro-edge on the generatrix (the crossing sectional area of cut) [mm^2^]. The shear coefficients and the blade force coefficients can be obtained by calibration methods [30,31,32]. However, it is impossible to calibrate the coefficients of the mechanical model for each patient.

The micro-edge chip width db could be expressed in a function of the micro-axial position angle $d\theta_{m}$ and the radius of cutter *R*, as follows:(9)$$db=Rd\theta_{m}$$

The micro-edge length ds is written as
(10)$$ds=R\sqrt{1+\cos^{4}\theta_{m}\tan^{2}\beta_{G}}\;\cdot \;d\theta_{m}$$

The thickness of the undeformed chip $t_{n}$ is the projection of feed per tooth in the normal direction of the sphere. According to the theory of Merdol and Altintas [33], the thickness of the undeformed chip $t_{n}$ is shown as the equation:(11)$$t_{n}=\frac{v_{t}\;\cdot \;n_{t}}{\left|n_{t}\right|}$$
where $v_{t}$ denotes the projection of the feed rate per teeth of the cutter on the rotating milling cutter coordinate system (MCCS). $n_{t}$ denotes a vector from the centroid of the ball end to the micro-edge, and $\left|n_{t}\right|=R$. The feed rate per tooth is written as:(12)vt=60·vdin·N
where *N* is the self-spinning speed of the cutter [RPM], *n* is the number of flutes on the cutter, and vdi is the tool feed data in the *i*-MCS [mm/s].

The cutting forces at the micro-edge are not parallel to the coordinate axes of the MCCS, which is modeled by the following equation:(13)$$\left(\begin{array}{c} dF_{xT}\\ dF_{yT}\\ dF_{zT} \end{array}\right)=\left(\begin{array}{ccc} -c\theta_{m}c\varphi_{m} & -s\theta_{m}c\varphi_{m} & s\varphi_{m}\\ -c\theta_{m}s\varphi_{m} & -s\theta_{m}s\varphi_{m} & -c\varphi_{m}\\ -s\theta_{m} & c\theta_{m} & 0 \end{array}\right)\left(\begin{array}{c} dF_{r}\\ dF_{a}\\ dF_{t} \end{array}\right)$$
where *s* and *c* denote the trigonometric functions of sin() and cos(), respectively. $dF_{xT}$, $dF_{yT}$, and $dF_{zT}$ are the components of the cutting force on the coordinate axes. $\varphi_{m}$ is the circumferential angle of the micro-edge, which is expressed as:(14)$$\varphi_{m}=\varphi_{0}-\sin\theta_{m}\tan\beta_{G}$$

Based on the finite element method, the instantaneous milling forces of the cutter in the MCCS could be obtained as follows:(15)$$\left\{\begin{array}{c} F_{xT}=\sum dF_{xT}\left(P_{i}\right)\\ F_{yT}=\sum dF_{yT}\left(P_{i}\right)\\ F_{zT}=\sum dF_{zT}\left(P_{i}\right) \end{array}\right.$$
where $F_{xT}$, $F_{yT}$, and $F_{zT}$ are the components of the cutting force.

#### 2.2.5. CT Image–Force Mapping Model

During the bone cutting process, the cutter faces variations in bone shapes and bone material properties. The above mechanical model considers many factors but still lacks consideration of the time-varying material property. Some works have shown that bone materials’ strength and modulus have a power relationship with density [27], and the bone density is proportional to the value in a CT image [26]. Thus, the CT image information is introduced into the bone milling model herein.

The cutting forces are weighted by the power of the grayscale value of the CT images. This is also consistent with the viewpoint of the literature [25]. **Equation** (Equation 16) reveals the weighted personalized bone milling forces.
(16)$$\left\{\begin{array}{c} dF_{r}=\lambda^{\mu }\;\cdot \;\left(K_{rc}\;\cdot \;t_{n}\;\cdot \;db+K_{re}\;\cdot \;ds\right)\\ dF_{a}=\lambda^{\mu }\;\cdot \;\left(K_{ac}\;\cdot \;t_{n}\;\cdot \;db+K_{ae}\;\cdot \;ds\right)\\ dF_{t}=\lambda^{\mu }\;\cdot \;\left(K_{tc}\;\cdot \;t_{n}\;\cdot \;db+K_{te}\;\cdot \;ds\right) \end{array}\right.$$
where λ is the image grayscale value at the infinitesimal element’s position and μ is the power relationship coefficient.

In order to eliminate the gray shift due to changes in CT devices’ performance, the image grayscale value extracted is normalized herein. In **Equation** (Equation 17), a piecewise linear function is used as the normalization rule.
(17)$$\lambda =\left\{\begin{array}{cc} 0, & x\leq m_{\min},\\ \frac{1}{m_{\max}-m_{\min}}\left(x-m_{\min}\right), & m_{\min}<x<m_{\max},\\ 1, & x\geq m_{\max}. \end{array}\right.$$
where *x* represents the grayscale value of a CT image voxel. $m_{\min}$ and $m_{\max}$ represent normalized parameters, which can be determined according to the gray distribution of the specified CT images. Typically, a bone contains both cortical bone and cancellous bone, $m_{\min}$ can be taken as the mean CT value of the cancellous bone area minus two times the standard deviation, and $m_{\max}$ is the mean CT value of the cortical bone area plus two times the standard deviation [25].

### 2.3. Tip Position Estimation

The local density distribution of the vertebrae can be used as the identifiable localization feature [28,34]. However, it is difficult to directly obtain real-time bone density during surgery. According to **Equation** (Equation 16), the fluctuation of the milling signals consists of information including the object density. In the clinical environment, many factors can introduce noise into the intra-operative signals, which affects the estimation of the milling states.

In the proposed method, the estimation principle of the tip position is to search the index of the maximum similarity between the milling prediction sequences and the real-time force signal. To improve the accuracy, the fitting methods are used for estimation of the tip position. As shown in Figure 7, the planned workspace is digitized into paralleled components subject to the planned paths. Each component is further divided into continuous grids. Each grid consists of several CT image voxels in series. The image voxel works as the basic metric unit in this work. According to the CT image–force model, each grid can be transformed into one data sequence with the same length.

Every prediction sequence $g\left(i,j,k\right)$ can be regarded as the unique landmark indicating the position $P_{p}\left(i,j,k\right)$ of the *k*-th segment in the *j*-th path of the *i*-th layer inside the workspace of the surgical task, as shown in **Equation** (Equation 18).
(18)$$g\left(i,j,k\right)\Rightarrow P_{p}\left(i,j,k\right)$$

The intra-operative real-time force signal is segmented into continuous sequences with the length of $n_{s}$ by a tumbling window. When the sampling frequency of the sensor is $f_{s}$ [Hz], the physical length *l* [mm] of a signal sequence can be expressed as:(19)$$l={v_{d}n_{s}/f\;}_{s}$$

The initial phase of the milling cutter has an influence on the milling signal. It can be ignored when the milling parameters agree with the following relationship:(20)$$\frac{60}{n\;\cdot \;N}v_{d}\ll \Delta x_{pix}$$
where $\Delta x_{pix}$ represents the physical size of one voxel of the CT images. It means that the ratio of the self-spinning speed of the cutter to the feed rate should be high enough.

Similarity mapping between the prediction sequences and the force signal segment enables estimation of the tip position. Due to the influence of bone motions, sensor noise, and signal synchronization, it is a great challenge to find precious paired sequences. To address this problem, the dynamic time warping (DTW) algorithm is employed to improve robustness.

Dynamic time warping (DTW) is a well-known machine learning method to measure the similarity of two temporal sequences, allowing similar shapes to match even if they are out of phase in the time axis [35,36,37]. It means that the sequences are warped in a nonlinear fashion to match each other. The steps of the position estimation method are as follows:Calculate the similarity metric $M(S_{rm})$. $M(S_{rm})$ is a three-dimension matrix that is the optical DTW distance of the signal segment $S_{rm}$ and the predicted sequences. Its component $d_{i,j,k}$ in the metric $M(S_{rm})$ represents the similarity with the predicted sequence of the index I[i,j,k], which can be expressed as:
(21)$$d_{i,j,k}\left(S_{rm}\right)=DTW\left(S_{rm},S_{i,j,k}\right)$$
where $1\leq i\leq N_{l}$, $1\leq j\leq N_{traj}$, and $1\leq k\leq N_{s}$. $N_{r}$, $N_{traj}$, and $N_{l}$ are the number of grids on one path, the number of paths on every layer, and the number of layers in the planning task, respectively.The optical DTW distance is used to represent the similarity between two sequences, as follows:
(22)$$d_{i,j,k}=C\left(N_{s},N_{s}\right)$$
where $C\left(N_{s},N_{s}\right)$ denotes the component value of the index $\left(N_{s},N_{s}\right)$ in the DTW distance matrix *C*.The optical DTW distance between two sequences is derived based on the principle of the shortest distance, as follows:
(23)$$C\left(i,j\right)=D\left(i,j\right)+\min\left\{\begin{array}{cc} \; & C\left(i-1,j\right)\\ \; & C\left(i,j-1\right)\\ \; & C\left(i-1,j-1\right) \end{array}\right.$$
where $1\leq i\leq N_{s}$, $1\leq j\leq N_{s}$, and *D* is the distance matrix. In this work, we use the Euclidean metric to obtain the distance matrix *D*, as follows:
(24)$$D\left(i,j\right)=\left|s_{rm}\left(i\right)-s_{i,j,k}\left(j\right)\right|$$
where $s_{rm}\left(i\right)$ indicates the *i*-th data point in the real-time sequence $S_{rm}$, and $s_{i,j,k}\left(j\right)$ indicates the *j*-th data point in the prediction sequences.Find the index of the sequence with the highest similarity as the paired coordinate position:
(25)$$I*=\arg\min_{I_{i,j,k}}\;\left(M\right)$$The tip position is obtained according to **Equation** (Equation 18) as
(26)$$\hat{P}=P\left(I*\right)$$

### 2.4. Optimization on the Position Estimation

By a fitting method, points of the real-time position of the cutting tip can be used to derive the location of the milling path. The schematic view of estimating on the real-time milling path is as shown in Figure 8. We adopt the random sample consensus (RANSAC) algorithm as the fitting method to estimate the milling path. This algorithm was proposed by Fishler and Bolles in 1981, and it is a widely used robust estimator [38].

The RANSAC algorithm for fitting the milling path is shown in Algorithm 1. In this step, $n_{1}$ is 2, $k_{1}$ is 2000, and $t_{1}$ is 1.0.

It is assumed that the deviation of every milled path from those planned is identical. During the milling operation, the space formed by the continued milled paths agrees with some geometric conditions. These are used as the geometric constraints to estimate the location of the milled space, as well as to evaluate the deviation from the planned paths. We also use the RANSAC algorithm to obtain the milling depth, and deviation of the milled part, as shown in Algorithm 2. In this step, $n_{2}$ is 30, $k_{2}$ is 2000, and $t_{2}$ is 1.0.
**Algorithm 1** RANSAC for fitting the milling path.Input:
     data—The matched coordinate point $P_{t}$ on the current milling path, as well as the milled path.
     fitting model—The model for fitting a 3D line:
(27)$$\frac{x-x_{0}}{l}=\frac{x-y_{0}}{m}=\frac{x-z_{0}}{n}$$
where the vector (l,m,n) indicates the direction of the fitted line in the 3D space, $\sqrt{l^{2}+m^{2}+n^{2}}=1$.
     evaluation model—the deviation distance from the data point to the fitting line:
(28)$$\varepsilon =\frac{\left(P_{i}-P_{0}\right)\;\cdot \;\left(v_{0}\times \left(P_{i}-P_{0}\right)\times v_{0}\right)}{∥P_{i}-P_{0}∥}$$
where $P_{0}=\left(x_{0},y_{0},z_{0}\right)$ and $v_{0}$ indicate one point on the fitted line, and the line direction, respectively.
     $n_{1}$—BestNum: Minimum number of data points to estimate model parameters.
     $k_{1}$—Maximum number of iterations allowed in the algorithm.
     $t_{1}$—The threshold value to determine data points that fit well by the above models.
Output:
     bestFit—The parameters for the fitting line: l,m,n


**Algorithm 2** RANSAC for estimating the milling depth.Input
    data—The fitting lines of milled paths.
    Fitting model—The model for fitting a displace deviation between the planned space and the current milled space:
(29)$$\left\{\begin{array}{cc} \; & \Delta n_{r}=E\left(R_{i,j}-r_{i,j}\right)\;\\ \; & \Delta n_{l}=E\left(L_{i,j}-l_{i,j}\right) \end{array}\right.$$
where $\Delta n_{r}$, and $\Delta n_{l}$ represent the deviations on the *y*-axis and the *z*-axis of the WCS, respectively. $E\left(\;\cdot \;\right)$ represent the mean function. $R_{i,j}$, and $r_{i,j}$ indicate the *y*-axis components of the estimation and planning value of a path. $L_{i,j}$, and $l_{i,j}$ indicate the *z*-axis component of the estimation and planning value of a path.
     Evaluation model—The geometric constrain, as:
(30)$$\varepsilon_{s}=\sqrt{{\left(R_{i,j}-r_{i,j}-\Delta n_{r}\right)}^{2}+{\left(L_{i,j}-l_{i,j}-\Delta n_{l}\right)}^{2}}$$
     $n_{2}$—BestNum: Minimum number of data points to estimate model parameters.
     $k_{2}$—Maximum number of iterations allowed in the algorithm.
     $t_{2}$—The threshold value to determine data points that fit well by the model.
Output:
     bestFit—The deviation between the planning workspace and the actual milled workspace.


## 3. Experiments and Results

### 3.1. Simulations of the CT Image–Force Model

In order to depict the milling force to various densities, we built a virtual CT image with the stepped pixel gray level as the milling object. The size was 12.0 mm × 12.0 mm × 6.0 mm. The physical dimension of each voxel was defined as 0.25 mm × 0.25 mm × 0.25 mm. As shown in Figure 9, the grayscale in the image was divided into 10 step intervals from 0 to 1400. This range of grayscale conformed to that of the common CT images. Following the above definition of the WCS, a coordinate system was built, where the x-axis was defined by the gradient direction in this CT image.

It was assumed that the milling feed rate was 0.5 mm/s, the grinding speed was 800 RPM, the inclined angle of the cutter was 30°, the helix angle of the chisel edge was 30°, the one-layer milling depth was 0.8 mm, the number of the flutter was 4, the distance from the edge *h* was 0.25 mm, and the step angle of the axial angel of cutting edges was 4°. The sampling frequency of the cutting force sensor was 20 Hz. In this simulation, the milling parameters and model parameters are listed in Table 1.

We adopt the parameters in Table 2 for the force coefficients of **Equation** (Equation 16). In addition, it was assumed that the initial phase of the cutter on each milling path was random. Then, we obtained the milling forces $F_{x}$, $F_{y}$, $F_{z}$, and $M_{z}$ in the *i*-MCCS, as shown in Figure 10. The instantaneous gray was quantified by the average of the gray value of image voxels contacting with the valid micro-edges on the CWE area.

### 3.2. Signal Analysis on the Laminar Milling

According to studies [39,40], although the bone strength of humans is different from that of animals, the structural and functional properties of bone tissue are conserved. Animal experiments serve as the foundations for human trials in the field of bone tissue injury and repair [41]. Therefore, three groups of robot-assisted laminar milling experiments were conducted to verify the relationship between the bone density distribution and the milling force signal. Of these, two groups were conducted on porcine thoracic and lumbar laminas, respectively. The third group of experiments was on the plastic lumbar made by 3D printing technology (Material: R4600, Manufacturer: Wenext Technology Co., Ltd., Shenzhen, China). Different from porcine laminas, the plastic is profiled with uniform density. Every group of experiments was repeated three times. The milling force signal was collected during the operations. Through the spatial transformation as shown in **Equation** (Equation 31), we can obtain the milling force in the *i*-MCCS through the force/torque sensor.
(31)Fm=Adfm·Ff
where Adfm is the adjoint transformation matrix from the robotic flange coordinate system to the *i*-MCCS, and Ff represents the force/torque data in the format of the vector ${\left(f_{x},f_{y},f_{z};m_{x},m_{y},m_{z}\right)}^{T}$.

Following the proposed milling strategy, a UR 5e robot arm with a six-axis force/torque sensor was used to assist the milling operation, as shown in Figure 11a. During the milling operations, the milling tool was constrained by the guide as the end effector of the robot arm. The spinning speed of the cutter was manually controlled. In these experiments, we adopt the bi-direction milling during the bone milling. The path planning is listed in Table 3, where $n_{pos}$ indicates the number of paths in the positive direction, $n_{neg}$ indicates the number of paths in the negative direction, and $n_{data}$ indicates the length of every data sequence. Moreover, the milling depth was 0.6 mm, and the feed rate was about 1 mm/s. It was noticed that the cutter did not contact the object during the first-layer milling procedure.

Correlation analysis was used to determine the relation between the milling force signal and bone density. As shown in **Equation** (Equation 32), we used the Pearson correlation coefficient to quantify the similarity among the milling sequences subject to different paths on the same object. The result can be written by a matrix. The correlation matrix was shown in the format of a figure, as shown in Figure 12. The diagonal elements in the correlation matrix represented the autocorrelation coefficient, and all of them were scaled to 1.0. If the correlation coefficient of two sequences was below 0.4, their relationship was regarded to be independent.
(32)$$R\left(i,j\right)=corr\left(s_{i},s_{j}\right)$$
where $1\leq i\leq n_{seq}$, and $1\leq j\leq n_{seq}$. $n_{seq}$ was the sum of the positive direction paths and the negative direction paths. $s_{i}$ and $s_{j}$ were the *i*-th and *j*-th sequences.

### 3.3. Functional Simulations of the Position Estimation

The commercial software MATLAB R2021b was used to conduct the functional simulation of the proposed tip position estimation on a computer (CPU: Intel i5-1135G7, RAM: 16.0 GB). Following the anatomical model in the commercial software Complete Anatomy [42], we created the segmentation annotation on the lamina in CT images using the open-source software ITK-SNAP [43]. An open-source library MNI2FS (MNI2FS: high-resolution surface rendering of MNI registered volumes) in MATLAB was used to extract the bone structure information, segmentation, and image information from the medical image files. In MATLAB, the point cloud method was used to demonstrate the segmented lamina and vertebra. The segmented lamina was marked by a frame. Further, we manually made the plan of the milling paths, as shown in Figure 13.

Milling efficiency is affected by the direction of feed motion. Thus, we employed the milling strategy to ensure that valid milling interactions occur during the motion along the positive direction. The milling parameters in Table 1 and the force coefficients in Table 2 were also utilized to generate the milling prediction sequences. Several factors, like bone motions, sensor noises, and unstable motions of a cutting tool, can introduce disturbance. Referring to the literature [7,44,45,46], it was assumed that the deviations in the simulated milling paths were limited and random. Moreover, the unstable feed rate of the milling cutter was modeled in a sine function.

In this simulation, nine sets of CT images with lumbar vertebra were randomly chosen from the VerSe dataset [47], including verse 503, 507, 620, 704, 714, 718, 756, 762, and 835. These cases and the CT images were from different centers. Every vertebra was profiled with different shapes, as well as the size of the voxel in the CT images being different. We chose randomly from lumbar vertebras in every case. Correspondingly, the surgical paths were different, as shown in Table 4.

The estimation deviation can be divided into two parts, including the distance deviation and the milling depth deviation. Through a comparison of the similarity calculations, we verified the validity of the proposed estimation. The Pearson correlation was used to take the place of the DTW algorithm. The length of the data sequence was 15 herein. The errors of the tip position estimations are shown in Figure 14. The estimation error based on the correction coefficient was larger than that based on the optical DTW distance.

In the case of VerSe 762, the left lamina inside the second lumbar vertebra was narrow, so we only planned two paths on every layer, and 11 layers. In this group of simulations, the number of data for the fitting model was re-defined by eight in Algorithm 2. The initial deviations are listed in Table 5. Correspondingly, the estimation errors are shown in Figure 15. The means of the estimation error were all less than 1 mm.

In the case of VerSe 620, we chose the left lamina of the second lumbar vertebra as the target. In this case, we planned 12 milling layers, and four paths on every layer. The path planning and the pre-defined deviations are listed in Table 4. The corresponding milling predictions are shown in Appendix B. When the BestNum in Algorithm 2 was 30, the estimation errors based on the correlation coefficient and the DTW distance are shown in Appendix C.

We can estimate the tip position when the number of milled paths was more than the BestNum in Algorithm 2. Furthermore, when the BestNum values were 20, 25, 30, and 32, the estimation errors corresponding to various conditions are shown in Figure 16.

Due to bone motion and the deformation of the cutter, the feed rate was unstable during the operation. We used a sine function to model the unstable feed rate. With the amplitudes of 0.05 mm/s, 0.10 mm/s, 0.15 mm/s, 0.20 mm/s, and 0.30 mm/s, five groups of 0.5 mm/s feed rates were used. The initial phase of the milling cutter, as well as the deviation, was also random. Subject to different amplitudes, the estimation errors are shown in Figure 17. The results showed that the estimation error was less than 1.0 mm, and the error based on the DTW algorithm was less than 0.5 mm.

## 4. Discussion

In other works, the principle of state recognition for spinal surgery is to detect the signal relieved when the cutter breaks through the second bone layer. As mentioned earlier, due to the difference in the bone properties of various specimens, the threshold for identified bone states is different. The focus of this paper is to address this problem. We take advantage of intra-operative CT images to predict the milling states. Then, we perform position estimation based on the similarity mapping between the predicted and real-time force signals by using the machine learning algorithm and the iterative algorithm. Therefore, we do not employ neural networks or deep learning algorithms in this work. Thus, the proposed method is not dependent on data samples, and does not require extensive training.

In the proposed CT image–force model, the force coefficients in **Equation** (Equation 16) should be obtained by a complex calibration process [23,30], whereas it is impossible to predict the milling force without loss of accuracy. In this work, through the prior knowledge gained from the CT image, we prefer to obtain the time-domain characteristics of milling forces rather than precious results. Moreover, the principle of position estimation is based on the similarity mapping of time-domain signal features. Various groups of CT images from the VerSe dataset were utilized to verify the validation of the proposed estimation method. The resolutions of CT images from different centers and CT devices are different. The results showed that the estimation method can adapt to CT images from different sources and CT devices.

To eliminate the influences of uncertain factors, such as bone motion, signal noise, signal synchronization, and so on, we make use of the DTW algorithm in the proposed framework. Compared with the linear algorithm, the usage of the DTW algorithm can improve estimation accuracy. The results showed that estimation accuracy can be up to the sub-millimeter level. However, the estimation is also affected by the step distances between the planning paths.

Time efficiency will degenerate with the increase in the milling volume of the surgical task, owing to the workload of the mapping calculation. In addition, time efficiency is affected by the length of the segmented data sequences. When the length is too long, the noise can affect the accuracy of the similarity matching. When the length is too short, the amount of location information is too little to generate the effective feature. Currently, application of the proposed framework is limited by the time-efficient performance.

## 5. Conclusions

This work has established a framework for state estimation of the intra-operative tip position during laminar grinding. Based on our proposed CT image–force model, we have demonstrated that prior knowledge of milling states, subject to specific surgical planning, can be extracted from pre-operative CT images. By integrating real-time milling force signals with this model, we have successfully estimated the tip position with improved robustness and accuracy, as evidenced by our simulation results. The DTW algorithm and the RANSCA algorithm have been effectively integrated to enhance performance; the estimation error is less than 1 mm, achieving an accuracy level up to the sub-millimeter. We employ many cases with different image resolutions from the VerSe dataset to verify that the proposed method is robust.

In future, our focus will shift towards optimizing the time efficiency of our framework from aspects of estimation algorithms. We plan to conduct laminar grinding experiments to empirically validate the proposed framework, aiming to assess and refine its performance under clinical conditions.

## Figures and Tables

**Figure 1 sensors-25-00238-f001:**
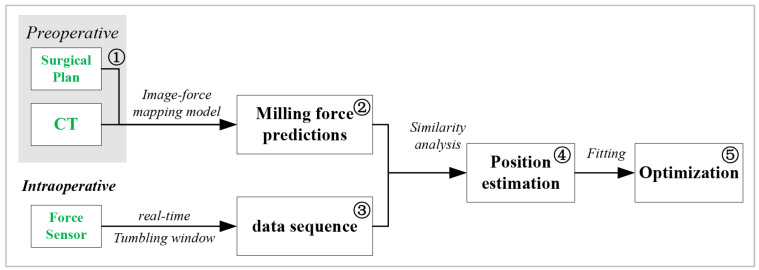
Framework of state estimation on laminar grinding.

**Figure 2 sensors-25-00238-f002:**
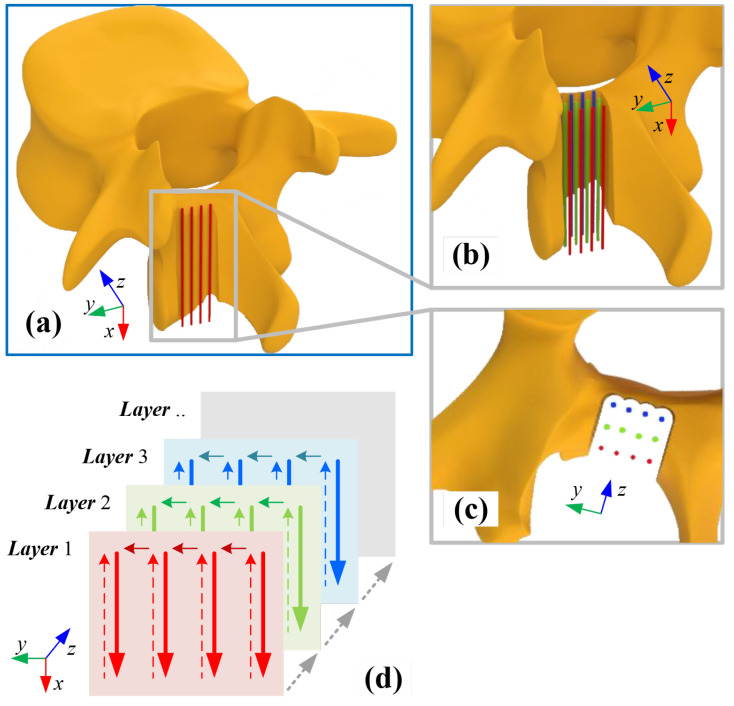
Schematic view of the layer-by-layer policy. (**a**) Schematic view of the thinned lamina, where four red lines represent the paths on the first milling layer. (**b**) Three groups of color lines represent the different layers, respectively. (**c**) Top view of the milling paths. (**d**) Spatial motions of the cutting tip during laminar grinding.

**Figure 3 sensors-25-00238-f003:**
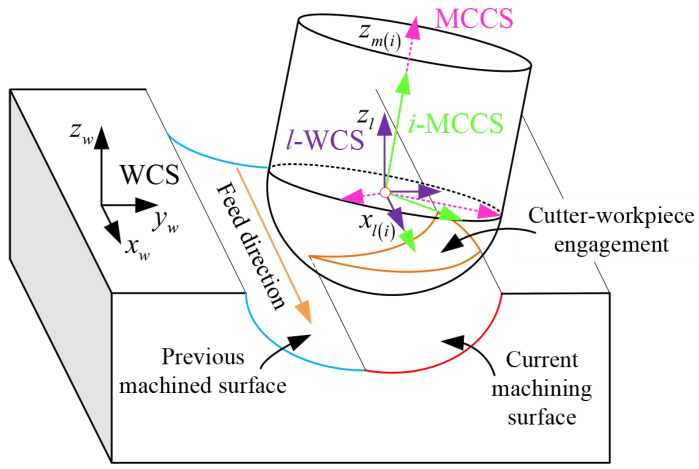
Schematic view of the bone milling process.

**Figure 4 sensors-25-00238-f004:**
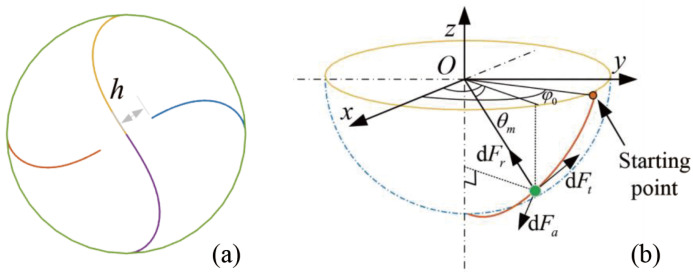
Geometric model of a four-flute ball-end cutter. (**a**) The bottom view of a four-flute cutter. (**b**) Isometric view of one cutting edge on a ball-end cutter.

**Figure 5 sensors-25-00238-f005:**
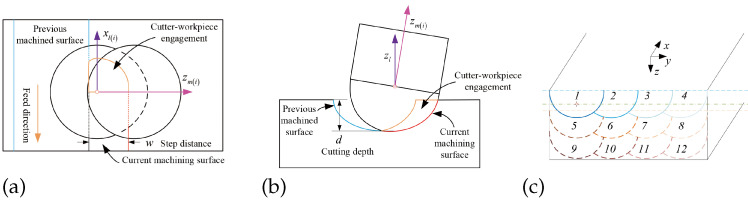
Schematic view of the cutter–workpiece engagement (CWE). (**a**) Top view. (**b**) Side View. (**c**) View of the CWE of several paths in the layer-by-layer policy.

**Figure 6 sensors-25-00238-f006:**
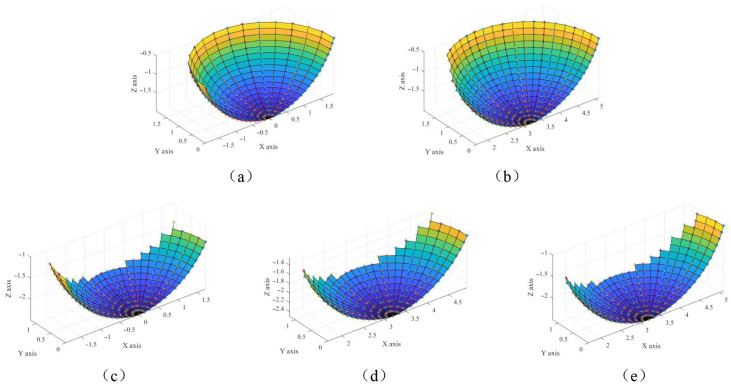
Schematic view of the CWE areas of different milling paths. (**a**) Valid CWE area of the first path on the top layer. (**b**) Valid CWE area of the others on the top layer. (**c**) Valid CWE area of the first path on the other layers. (**d**) Valid CWE area of the middle paths on the other layers. (**e**) Valid CWE area of the last path on the other layers.

**Figure 7 sensors-25-00238-f007:**
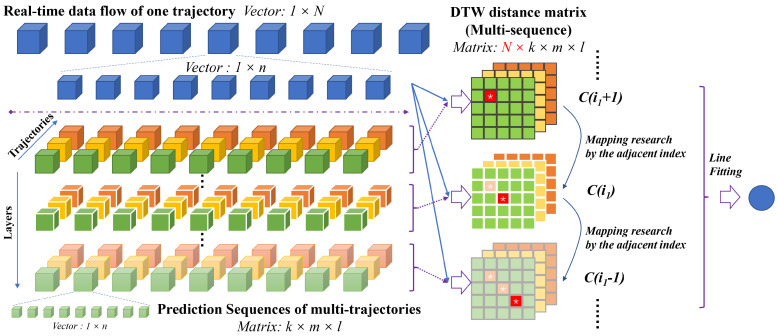
Tip position estimation by the method of similarity matching.

**Figure 8 sensors-25-00238-f008:**
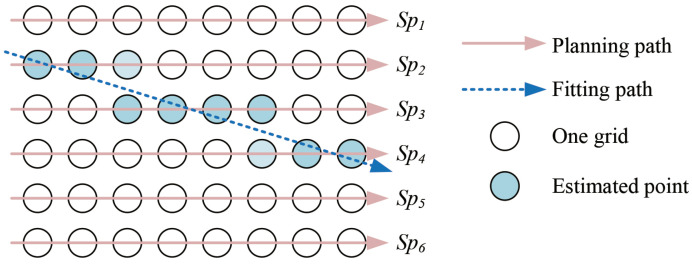
Schematic view of estimating the location of the milling path by a fitting algorithm. The blue circle indicates the estimated positions. The blue dashed line indicates the estimation on the current path. The red line indicates the planned paths.

**Figure 9 sensors-25-00238-f009:**
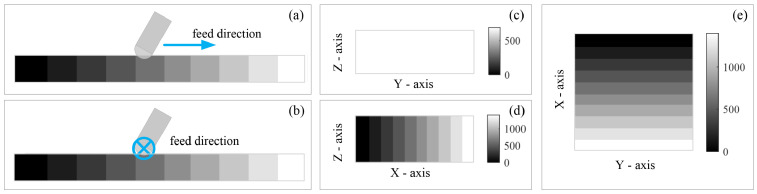
Schematic view of milling on one virtual object filled with gray gradient. (**a**) Simulation where milling paths paralleled the direction of the gray gradient, in which the color indicates its grayscale. (**b**) Simulation where milling paths ran vertical to the direction of the gray gradient, in which the color indicates its grayscale. (**c**) Front view of the virtual object. (**d**) Side view of the virtual object. (**e**) Top view of the virtual object.

**Figure 10 sensors-25-00238-f010:**
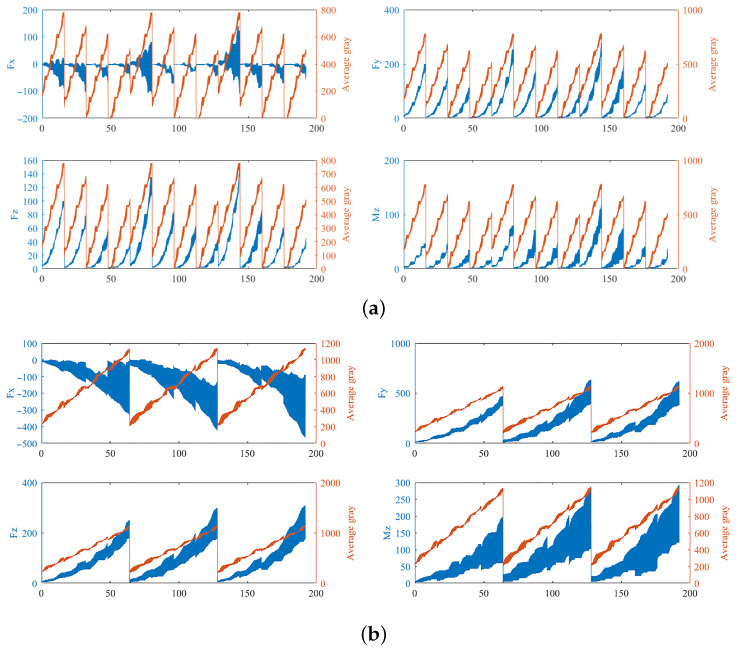
(**a**) Milling forces subject to the paths along the gray-gradient direction. (**b**) Milling force responding to the trajectories vertical to the gray gradient. The blue line depicts the prediction, while the red line depicts the gray average of the milling path in the CT image. The horizontal axis indicates the sequential order of the sampled predictions.

**Figure 11 sensors-25-00238-f011:**
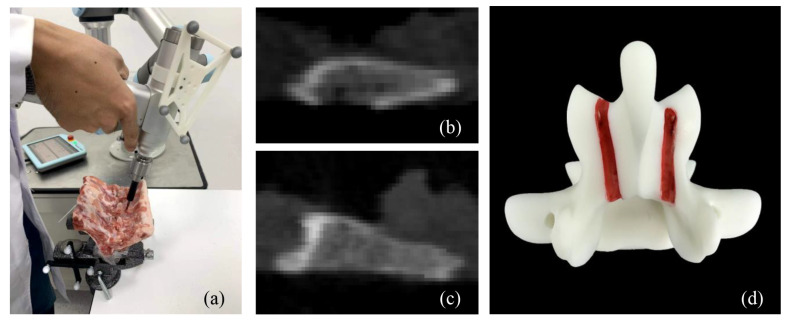
Experimental platform of the robot-assisted laminar milling and the experimental objects. (**a**) Robot-assisted milling operation. (**b**) CT image of porcine laminas. (**c**) CT image of porcine laminas. (**d**) The plastic model is made by 3D printing technology.

**Figure 12 sensors-25-00238-f012:**
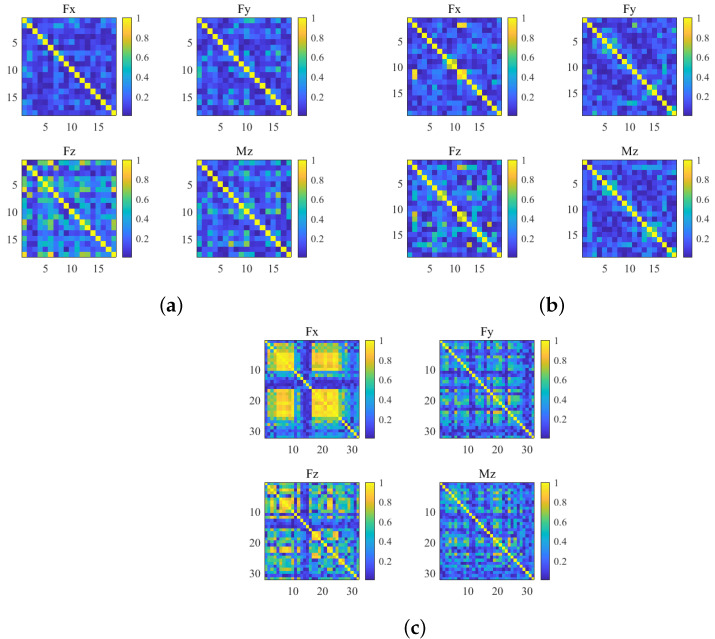
(**a**) Correlation relationship among the milling force sequences subject to the thoracic lamina. (**b**) Correlation relationship among the milling force sequence subject to the lumbar lamina. (**c**) Correlation relationship among the milling force sequence subject to the plastic model.

**Figure 13 sensors-25-00238-f013:**
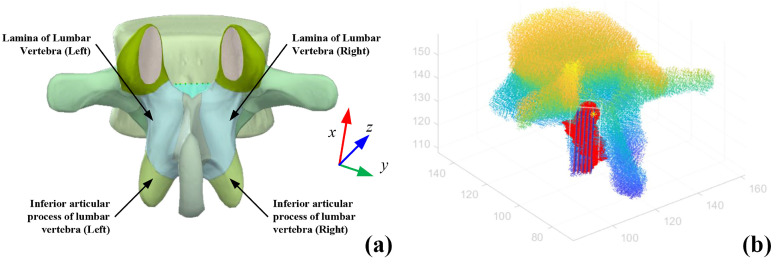
(**a**) Schematic view of the lumbar vertebra [42]. (**b**) The point cloud of a lumbar vertebra, where blue lines represent the planned paths, the red point represents the annotated lamina, and the yellow star represents the starting point of vertebra grinding. The color of the point cloud of the spinal bone indicates high information.

**Figure 14 sensors-25-00238-f014:**
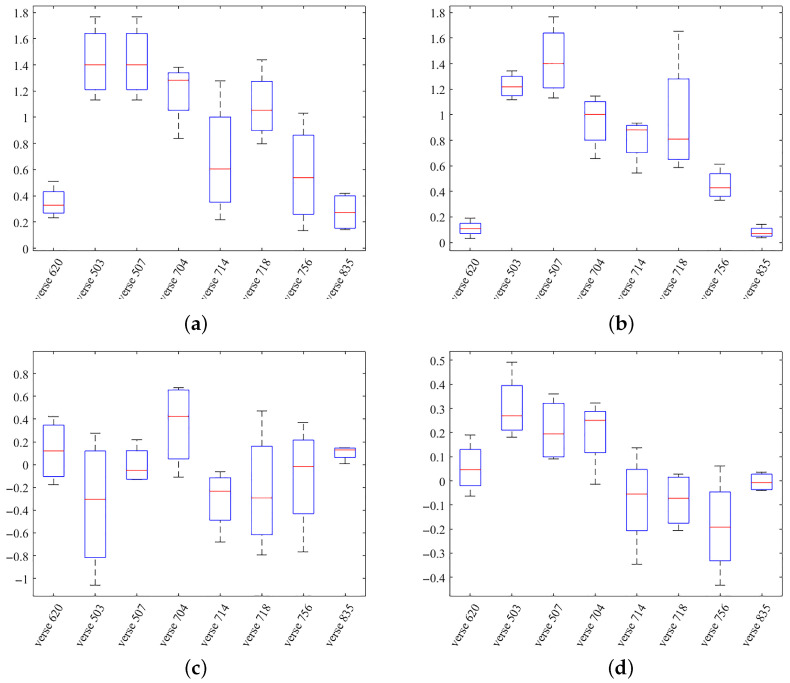
Estimation errors of the tip position. (**a**) Error of the distance deviation based on the correlation coefficient. (**b**) Error of the distance deviation based on the DTW distance. (**c**) Errors of milling depth estimation based on the correlation coefficient. (**d**) Errors of milling depth estimation based on the DTW distance. The unit of errors is millimeters.

**Figure 15 sensors-25-00238-f015:**
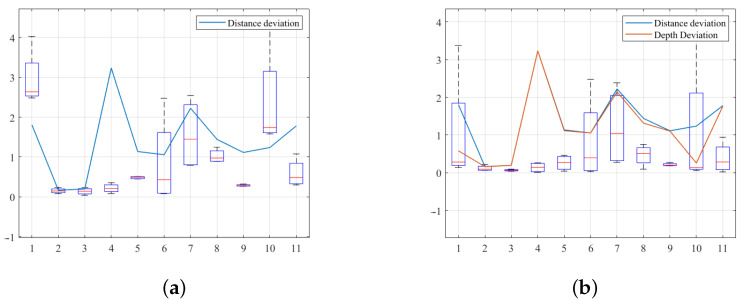
Estimation errors of simulations for Verse 762. (**a**) Absolute distance deviation. (**b**) Absolute milling depth deviation. The unit of the deviation is millimeters.

**Figure 16 sensors-25-00238-f016:**
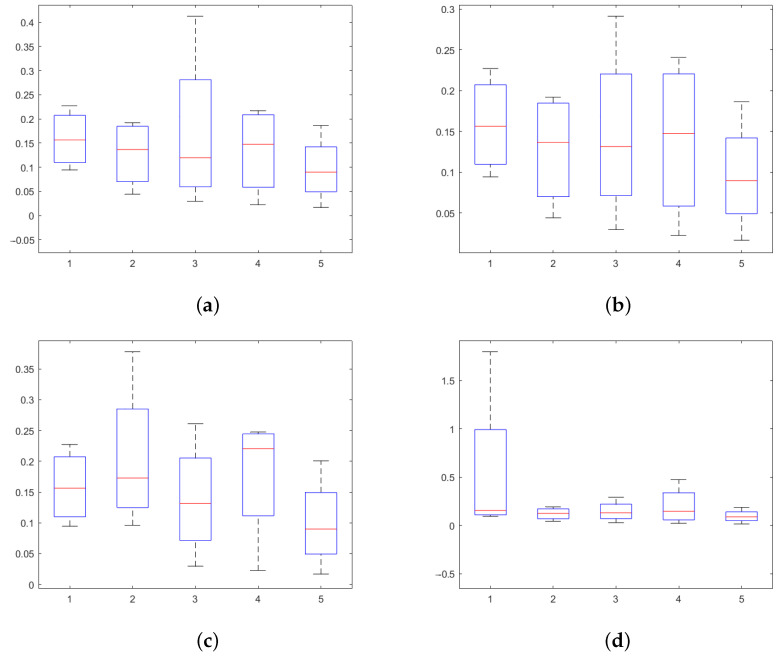
Estimation error in the case of Verse 620. (**a**) BestNum20. (**b**) BestNum25. (**c**) BestNum30. (**d**) BestNum32. The unit of the errors is millimeters.

**Figure 17 sensors-25-00238-f017:**
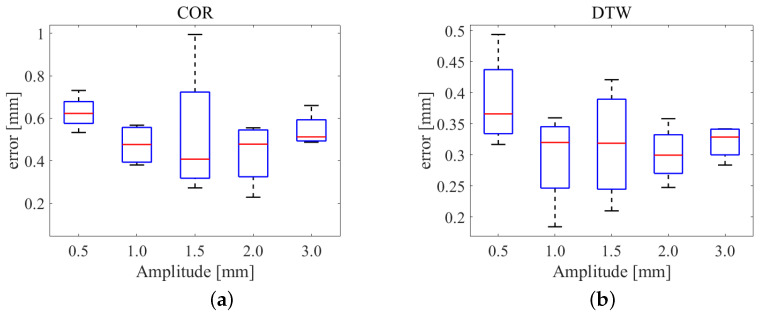
Estimation errors subject to various feed rates. (**a**) Estimation error based on Pierre’s relevant analysis. (**b**) Estimation error based on the DTW algorithm.

**Table 1 sensors-25-00238-t001:** Milling parameters in the CT image–force model simulations.

Parameters	Symbol	Value
Feed rate	$v_{d}$	0.5 mm/s
Spinning speed	*N*	800 RPM
Milling depth	*d*	0.8 mm
Step distance	*w*	2.8 mm
Number of the flutter	$n_{t}$	4
Radius of the cutter	$r_{t}$	2 mm
Helix angle of the cutter		30°
Inclined angle of the cutter		30°
Step angle of the axial angle	$d\theta_{m}$	4°
Distance between cutting edges	*h*	0.25 mm
Sampling frequency		20 Hz
Number of milling layers		3
Number of paths on every layer		4
Upper normalized parameter	$m_{\max}$	1400
Lower normalized parameter	$m_{\min}$	80

**Table 2 sensors-25-00238-t002:** Coefficients of the milling model.

Symbol	Value
$K_{re}$	−1203.1 N/mm^2^
$K_{ac}$	−105.2 N/mm^2^
$K_{tc}$	2142.1 N/mm^2^
$K_{te}$	−75 N/mm^2^
$K_{ae}$	22.4 N/mm^2^
$K_{te}$	−199.1 N/mm^2^
μ	1.815

**Table 3 sensors-25-00238-t003:** Plan for the milling experiments.

Objects	$n_{\mathrm{pos}}$	$n_{\mathrm{neg}}$	$n_{\mathrm{data}}$	Penetration?
Thoracic	9	9	67	Yes
Lumbar	10	9	42	Yes
Plastic	16	16	51	No

**Table 4 sensors-25-00238-t004:** Path planning and initial deviations for the simulation cases.

No.	Layers	Paths	Deviations (rad, rad, rad; mm, mm, mm)	Incline Angle (°)
503	16	4	0.014, 0.011, 0.011; −0.65, −1.23,−0.27	1.19
507	11	5	0.021, 0.001, 0.010; 0.003, 0.24, −0.24	1.34
620	12	4	0.007, 0.004, 0.003; −0.12, −0.02, −0.16	0.48
704	11	3	0.005, 0.021, 0.024; 1.80, 0.69, −0.40	1.87
714	9	4	0.004, 0.013, 0.029; −0.67, −1.47, 1.17	1.86
718	10	5	0.060, 0.018, 0.007; −1.64, −1.87, 0.37	3.58
756	11	4	0.005, 0.013, 0.025; 0.65, −0.84, 1.26	1.65
835	15	7	0.026, 0.020, 0.015; 0.71, 0.09, 0.25	1.88

**Table 5 sensors-25-00238-t005:** Random deviation in the case of VerSe 762.

No.	Deviations	Incline Angle (°)
1	0.065, 0.024, 0.030; 0.43, 2.48, −0.24	4.35
2	0.054, 0.033, 0.024; −0.32, −0.01, 0.20	3.86
3	0.069, −0.000, 0.019; −0.35, 0.04, −0.21	4.11
4	0.053, 0.021, 0.022; 0.07, −0.18, −2.39	3.53
5	0.064, 0.010, 0.009; −1.43, 0.48, 0.97	3.78
6	0.055, 0.019, 0.022; 0.53, −0.02, 1.02	3.56
7	0.053, 0.029, 0.034; −1.24, −1.07, −1.43	3.97
8	0.009, 0.004, 0.004; 1.37, −0.89, −1.02	0.59
9	0.056, 0.031, 0.017; −0.77, 0.22, 1.19	3.77
10	0.064, 0.028, 0.029; 1.61, 1.74, 0.47	4.33
11	0.030, 0.003, 0.013; 1.36, −0.37, 1.43	1.89

## Data Availability

Data are contained within the article.

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
