# Peer review of "A Framework of State Estimation on Laminar Grinding Based on the CT Image–Force Model"

_sensors, 2025, doi:10.3390/s25010238_

Round 1
Reviewer 1 Report
Comments and Suggestions for Authors
This paper proposed a state estimation framework based on CT image-force, using preoperative CT images and intraoperative grinding force signals as the source inputs. According to such framework, a bone grinding force prediction model was established for surgical planning. The proposed framework integrates is reasonable and feasible. There are some issures as follows:
1. among 5 modules, the CT voxels extraction, the segmentatin of the milling force signal and fitting algorithms are all well studied. But CT image-force mapping model need more explanation, because CT image - grinding force mapping of different peoples vary dramatically during operation, even the robot position can influence the vibration forces. The authors are expected to explain why they can reliably obtain this relationship.
2. Because the cutting force. i.e., thrust force, toque, and lateral forces, vary with milling tool-bone contact angle and sharpness of cutter, how to mapping/identifying critical operation force in order to avoid injury?
3. network training need large amount of data, the authors are expected to explain the validity without enough data.
Author Response
Comment 1: among 5 modules, the CT voxels extraction, the segmentation of the milling force signal and fitting algorithms are all well studied. But CT image-force mapping model need more explanation, because CT image - grinding force mapping of different peoples vary dramatically during operation, even the robot position can influence the vibration forces. The authors are expected to explain why they can reliably obtain this relationship.
Response: Thanks for your comment. We revived the paper and explain some reasons. We had the same doubt before we finished this work. We found that, not only many factors affects prediction performance of the milling mechanical model, but also the prediction sequence compasses the effective time-domain signal features subject to the bone density distribution. Under the actions of various factors, they change nonlinearly in the time domain. Thus, we employ the DTW algorithm to address this problem.
Comment 2: Because the cutting force. i.e., thrust force, toque, and lateral forces, vary with milling tool-bone contact angle and sharpness of cutter, how to mapping/identifying critical operation force in order to avoid injury?
Response: The focus of this work is on estimating the position of the cutting tip rather than avoiding injury. In surgeries of the laminar grinding, the surgeons will not cut off spinal lamine. In my opinion, it is more important to estimate the position of the cutting tip due to lacking the effective sensing method. If we can control the cutting tip of the robot-assisted system, we can avoid injury.
Comment 3: network training need large amount of data, the authors are expected to explain the validity without enough data.
Response: In this work, we employ the machine learning algorithm and the iterative algorithm, rather than the network, or deep learning algorithms in this work. Different from other work, we do not use the networks to derive features from real-time time signals, so our proposed method do not need training data.

Reviewer 2 Report
Comments and Suggestions for Authors
This paper develops a state estimation method for based on the CT image-force model on the on the Laminar Grinding. The simulation results showed that the proposed method was effective when CT images were from different medical sources and devices. However, this work has nothing relationship with Sensors section.
Comments on the Quality of English Languageno comments
Author Response
Comment 1: This paper develops a state estimation method for based on the CT image-force model on the on the Laminar Grinding. The simulation results showed that the proposed method was effective when CT images were from different medical sources and devices. However, this work has nothing relationship with Sensors section.
Response: Thanks for your commend. The data processing and data fusion, and state estimation, are inside the aims & scope of Sensors. Actually, in our work, the CT image and the intro-operative milling force, as data sources, are fused to estimate the position of the cutting tip. The method of data processing is complex, so we explain the derivation of the data fusion and mechanical principle. In addition, the relative works had been accepted by Sensor, like State Recognition of Bone Drilling Based on Acoustic Emission in Pedicle Screw Operation.

Reviewer 3 Report
Comments and Suggestions for Authors
The objective of this study is to develop a framework for state estimation of the intra-operative tip position during laminar grinding. While the article presents a promising approach, it requires further revisions for enhanced clarity and precision.
The objective of this study is to develop a framework for state estimation of the intra-operative tip position during laminar grinding. While the article presents a promising approach, it requires further revisions for enhanced clarity and precision.
1. Line 70, please identify and summarise the limitations of the preceding study in this field of enquiry.
2. Figure 6, please specify the software used to create this image.
3. Figure 10, please clarify the meaning of the horizontal axis.
4. Line 386, please provide additional information that elucidates the notable advancements in methodology and comprehension vis-à-vis preceding literature and/or presents persuasively the potential for new applications thereof.
5. Part 4, a more detailed examination of the reasons why the methods proposed in this article are superior to others would be beneficial.
6. Line 417, it is recommended that the conclusion be rewritten and that quantitative results be included.
7. Line 27, “However, it is impossible, not only to sense deflection of a cutting bit, but also to obtain the position of the cutting tip.” Please add a reference here.
8. Line 54, “The pre-operative CT image has been used to predict the cutting state.” Please add a reference here.
9. Line 69, “the prior force information derived from the CT images does not seem to be consistent with actual measurements.” Please add a reference here.
10. Line 101, “In this work, because the cutter-workpiece engagement (CWE) is considered in the milling force model, we can ignore the influence of the milling strategy on the state estimation.” Could you please show the reason for this statement?
11. Line 100, you said that this work has 5 modules, and please indicate which part of 2.2-2.6 and 3.1-3.3 correspond to each module.
12. Line 100, Please explain in more detail the logic behind the arrangement of these 5 modules.
13. Figure 11, It seems that you used the bone of an animal to do the experiments, please indicate the reason and the similarity of the results that can be used for humans.
Author Response
Comment 1: Line 70, please identify and summarise the limitations of the preceding study in this field of enquiry.
Response: We have carefully considered your suggestion to discuss the limitations of our work in this part. We have revised the manuscript according to include a candid discussion of the limitations.
Comment 2: Figure 6, please specify the software used to create this image.
Response: They are created in the software Matlab. The functions includes mesh and surf. The code will be opened when the work is accepted.
Comment 3: Figure 10, please clarify the meaning of the horizontal axis.
Response: Thanks for your comment. The horizontal axis indicates the sequential order of the sampling points. We revived the paper.
Comment 4: Line 386, please provide additional information that elucidates the notable advancements in methodology and comprehension vis-à-vis preceding literature and/or presents persuasively the potential for new applications thereof.
Response: We revived this part. The advancements of this work are that our proposed method is robust, and it will be effective for various bone properties of different specimens.
Comment 5: Part 4, a more detailed examination of the reasons why the methods proposed in this article are superior to others would be beneficial.
Response: We revived this part. Different from others, we generate milling predictions as the prior knowledge, which dese not serve as the value of the called breakthrough signal. Whereas, we found the predictions compasses the time-domain signal characteristics. Some methods will be invalid due to changes of bone properties of various specimens, and nonlinear factors including operations. Thus, we employ the DTW algorithm to estimate the states by the time-domain signal matching.
Comment 6: Line 417, it is recommended that the conclusion be rewritten and that quantitative results be included.
Response: We revived the conclusion.
Comment 7: Line 27, “However, it is impossible, not only to sense deflection of a cutting bit, but also to obtain the position of the cutting tip.” Please add a reference here.
Response: We added the reference.
Comment 8: Line 54, “The pre-operative CT image has been used to predict the cutting state.” Please add a reference here.
Response: We added the reference.
Comment 9: Line 69, “the prior force information derived from the CT images does not seem to be consistent with actual measurements.” Please add a reference here.
Response: We added the reference.
Comment 10: Line 101, “In this work, because the cutter-workpiece engagement (CWE) is considered in the milling force model, we can ignore the influence of the milling strategy on the state estimation.” Could you please show the reason for this statement?
Response: The proposed milling strategy will help readers to understand the schematics of the laminal grinding. Actually, in the proposed milling force model, the cutting force is derived by cumulating the micro force at the micro edges. When the cutter-workpiece engagement is considered in the model, it means that the micro-edge will be active to generate the micro-soft when it is inside the CWE. Thus, we don’t think the milling strategy will affect its performance.
Comment 11: Line 100, you said that this work has 5 modules, and please indicate which part of 2.2-2.6 and 3.1-3.3 correspond to each module.
Response: We revived manuscript. The mapping relationship is proposed in Section 2.1.
Comment 12: Line 100, Please explain in more detail the logic behind the arrangement of these 5 modules.
Response: We added some details.
Comment 13: Figure 11, It seems that you used the bone of an animal to do the experiments, please indicate the reason and the similarity of the results that can be used for humans.
Response: As we all known, the animal experiments serve as the foundations for human trails in the field of bone tissue injury and repair. Moreover, referring to other works, we believe that the bone properties of human and animal bones are high-level similar in the field of the bone repair.

Round 2
Reviewer 2 Report
Comments and Suggestions for Authors
Actually, the data used in this work are not collected from the sensors, in the revised version, this work has nothing relationship with Sensors.
Reviewer 3 Report
Comments and Suggestions for Authors
I think this article can be accepted in its current form.